# Current and Future Insights for Optimizing Antithrombotic Therapy to Reduce the Burden of Cardiovascular Ischemic Events in Patients with Acute Coronary Syndrome

**DOI:** 10.3390/jcm11195605

**Published:** 2022-09-23

**Authors:** Abi Selvarajah, Anne H. Tavenier, Enrico Fabris, Maarten A. H. van Leeuwen, Renicus S. Hermanides

**Affiliations:** 1Department of Cardiology, Isala Hospital, 8025 AB Zwolle, The Netherlands; 2Cardiovascular Department, University of Trieste, 34127 Trieste, Italy

**Keywords:** acute coronary syndrome, P2Y_12_ inhibition, antithrombotic therapy

## Abstract

The pharmacological treatment strategies for acute coronary syndrome (ACS) in recent years are constantly evolving to develop more potent antithrombotic agents, as reflected by the introduction of more novel P2Y_12_ receptor inhibitors and anticoagulants to reduce the ischemic risk among ACS patients. Despite the substantial improvements in the current antithrombotic regimen, a noticeable number of ACS patients continue to experience ischemic events. Providing effective ischemic risk reduction while balancing bleeding risk remains a clinical challenge. This updated review discusses the currently approved and widely used antithrombotic agents and explores newer antithrombotic treatment strategies under development for the initial phase of ACS.

## 1. Introduction

Effective antithrombotic therapy is of paramount importance in reducing the burden of adverse ischemic events in patients with acute coronary syndrome (ACS) who undergo percutaneous coronary intervention (PCI) [1,2]. The synergetic antiplatelet effect exerted by oral dual antiplatelet agents (DAPT), consisting of aspirin and a P2Y_12_ receptor inhibitor, is the backbone of the current pharmacological treatment of ACS patients [3]. Various clinical trials established the benefit of DAPT in lowering the risk of recurrent ischemic events among such patients. Clopidogrel and the more potent ticagrelor and prasugrel are the guideline-recommended and most commonly used oral P2Y_12_ receptor inhibitors in ACS [4,5]. Despite the indisputable advances in their antiplatelet properties and the current evidence that oral DAPT mitigates the risk of stent thrombosis (ST), a noticeable number of ACS patients continue to experience adverse ischemic events. These detrimental events have been partly attributed to the variability in individual responsiveness to clopidogrel [6] and the unfortunate characteristic of all oral antiplatelet agents requiring 2–8 h to reach their maximum antiplatelet effect [7].

The role of intravenous (IV) glycoprotein IIb/IIIa inhibitors (GPIs) as antiplatelet agents was explored in numerous clinical studies in the setting of ACS to overcome the limitations of oral antiplatelet agents and was shown to reduce early adverse ischemic events [8,9,10,11]. However, the risk reduction by GPIs came at the cost of an increased risk of bleeding and thrombocytopenia, which led to a decline in the use of these parenteral agents in routine clinical practice and mainly limited them to bailout indications in cases of a no-reflow phenomenon or peri-procedural thrombotic complications [12,13,14,15,16].

The shortcomings of the current antithrombotic strategies in ACS patients prompted the development and substantial improvement of several classes of antithrombotic agents with different pharmacokinetic and pharmacodynamic characteristics, which have led to rapid changes in recommendations for their usage in the clinical practice of ACS in the last few decades. The ultimate goal is to aim for an early invasive approach by developing a highly intensive antithrombotic agent with rapid on- and offset to bridge the initial gap of delayed platelet inhibition in the acute phase of myocardial infarction (MI) and to help alleviate the recurrence of ischemic events and the burden of bleeding.

In this comprehensive and updated review, we present the currently approved and still widely used antithrombotic regimen and explore advent antithrombotic treatment strategies under development for the initial phase of ACS. The antithrombotic agents discussed in this review and their mechanisms of action are summarized in Figure 1 and Table 1.

## 2. Early Initiation of Antithrombotic Agents for Optimal Platelet Inhibition in Acute Coronary Syndrome

Early prompt revascularization of the infarct-related artery (IRA) is known to limit myocardial damage and improve the short- and long-term prognosis of ACS patients. The in-hospital mortality rate of ACS is decreased after the introduction of mechanical reperfusion with PCI and second-generation drug-eluting stents. Nevertheless, studies have shown worldwide variation in the provision of PCI for ACS patients due to delayed presentation and missing out on the PCI option. Zhang et al. showed that pretreatment with aspirin and a P2Y_12_ receptor inhibitor alone in the absence of reperfusion therapy significantly improved cardiac function, reduced cardiac fibrosis and inflammatory cell infiltration, and inhibited oxidative stress-induced platelet aggregation after MI in the mouse model compared to placebo [24]. These findings suggest that early pharmacological reperfusion with antiplatelet agents may have a protective effect on the development of adverse events and prognosis in ACS patients.

The importance of administration of antithrombotic therapy early after the presentation in STEMI patients undergoing PCI was first shown with the GPI abciximab [21]. The complementary benefit of prehospital treatment with abciximab alongside aspirin was associated with a higher rate of initial TIMI grade III flow pre-PCI with better myocardial salvage and improved clinical outcomes compared to placebo. However, routine prehospital treatment with GPI in NSTEMI (non-ST-segment elevation myocardial infarction) patients remains uncertain and has been associated with a higher risk of bleeding without a significant reduction in ischemic complications. A recent study showed a beneficial effect of routine GPI use in STEMI patients who received morphine to prevent the occurrence of ST, as morphine use is associated with diminished absorption of oral P2Y_12_ receptor inhibitors and delayed onset of such agents [25]. However, current guidelines do not universally recommend using these agents in NSTEMI patients.

Contrary to GPIs, various studies showed that prehospital administration of P2Y_12_ receptor inhibitor clopidogrel with a loading dose of 600 mg seems safe [18]. However, the early administration was not associated with improved patency of the IRA pre-PCI but showed a trend toward reducing adverse ischemic events post-PCI. The same applies to early treatment with the more potent ticagrelor; a loading dose of 180 mg ticagrelor was safe in STEMI patients planned for primary PCI but did not improve reperfusion of the IRA [19]. Early administration of crushed compared with integral prasugrel significantly improved platelet inhibition during the acute phase in STEMI patients undergoing primary PCI [20]. Though an improvement with TIMI III flow in the IRA pre-PCI or complete resolution of ST-segment deviations 1 h post-PCI was not observed, a considerable number of patients still exhibited inadequate platelet inhibition at the end of PCI.

Despite strong recommendations for using oral P2Y_12_ receptor inhibitors as soon as possible after the diagnosis of (N)STEMI, there is a paucity of data describing that these agents only provide beneficial effects after PCI. These findings suggest the need for alternative agents to achieve early patency of the IRA within a limited time window and to bridge the initial gap in platelet inhibition to reduce the infarction’s expansion and preserve left ventricular function.

Cangrelor, the only intravenous P2Y_12_ receptor inhibitor with rapid onset and offset of antiplatelet effect, may bridge the gap to optimal platelet inhibition by oral P2Y_12_ receptor inhibitors. However, it remains unknown whether prehospital treatment with cangrelor can improve coronary reperfusion pre-PCI as its efficacy and safety are not studied in prehospital settings [23,24]. Currently, the recommendation for cangrelor use is restricted to P2Y_12_-naive patients who undergo PCI only [14,15,16].

At present, antithrombotic agents for prehospital treatment in ACS patients under active investigation are the selective P2Y_12_ receptor antagonist developed for subcutaneous administration (Selatogrel) [26] as well as a subcutaneous GPI (Zalunfiban) [27].

## 3. Oral Antiplatelet Agents

### 3.1. Aspirin

The clinical use of aspirin was investigated in ISIS-2 (Second International Study of Infarct Survival), the first randomized clinical trial to demonstrate the efficacy and safety of aspirin in patients with suspected acute MI [28]. Treatment with aspirin significantly reduced total vascular mortality by 23% compared to placebo and was associated with a significant reduction in the rate of recurrent MI and stroke. Subsequently, aspirin became the pillar of ACS pharmacological treatment with the recommended routine administration of an IV loading dose of 320–500 mg, followed by an oral maintenance therapy unless contraindicated [14,15,16].

As multiple cellular pathways are involved in the activation and aggregation of platelets, and aspirin selectively targets COX (cyclooxygenase)-1 and TXA (Thromboxane A)-2, aspirin alone is ineffective in preventing platelet activation through the other active pathways in ACS [29,30]. Furthermore, aspirin resistance has been a topic of debate in the last decade in patients with coronary syndrome, which describes the recurrence of adverse cardiovascular (CV) events in patients on aspirin monotherapy with a lower-than-average platelet inhibition by aspirin [31]. Clinically, aspirin resistance is associated with a threefold higher risk of CV mortality, MI, and stroke [32]. These findings suggest that aspirin resistance is present; however, it is generally accepted that this issue is rare [33].

Hence, the current ACS guidelines recommend a loading dose of an oral P2Y_12_ receptor inhibitor alongside the aspirin, followed by 12 months of DAPT and lifelong monotherapy with aspirin thereafter, with daily maintenance dosages ranging from 75–100 mg (QD) in different countries. Higher maintenance dosages (≥160 mg) of aspirin are not preferred as they are associated with a higher risk of bleeding without a significant improvement in clinical ischemic events [34,35,36].

### 3.2. Clopidogrel

The second-generation P2Y_12_ receptor inhibitor, clopidogrel, belongs to the thienopyridine class of antiplatelet agents and exerts its antiplatelet effect by selectively and irreversibly preventing the binding of ADP to the P2Y_12_ receptor. Ticlopidine represented the first generation of P2Y_12_ receptor inhibitors with the same action as clopidogrel and was initially introduced to prevent thrombotic stroke. Despite its optimal antiplatelet effect with a maintenance dosage of 250 mg (BID), ticlopidine was associated with potentially fatal complications, including bone marrow suppression with neutropenia and thrombocytopenia [37]. Subsequently, clopidogrel, with a more favorable safety profile, is currently the most widely prescribed and used P2Y_12_ receptor inhibitor [38].

Clopidogrel is a prodrug that requires two-step metabolism to produce an active metabolite, which means it can take up to 2–4 h after a loading dose of 600 mg to achieve a significant inhibition of the P2Y_12_ receptors with a peak effect at 6 h [39]. Taking into account that only 15% of the absorbed clopidogrel undergoes metabolism to become an active metabolite, the remaining 85% of the absorbed clopidogrel is hydrolyzed by carboxylase to an inactive metabolite. Furthermore, patients with poor or intermediate metabolizer phenotypes due to genetic polymorphisms in CYP 450 enzymes (particularly CYP2C19) will have impaired active metabolite production, resulting in a lack of efficacy of clopidogrel [40,41]. However, genotype-guided selection did not significantly reduce adverse ischemic events compared to conventional clopidogrel therapy. Subsequently, 30–40% of patients on clopidogrel treatment retain high on-treatment platelet reactivity around PCI, which translates into long-term adverse CV events [6,42].

Consequently, newer oral P2Y_12_ receptor inhibitors, ticagrelor and prasugrel, emerged to achieve a more potent and homogeneous platelet inhibition with more predictable effects than clopidogrel. Nonetheless, combining clopidogrel and aspirin predominantly remains the cornerstone in acute treatment and secondary prevention of (recurrent) adverse ischemic events in ACS patients and patients undergoing elective PCI in real-world practice [43,44,45]. Moreover, for patients with a separate indication for oral anticoagulation, clopidogrel is recommended as part of DAPT alongside aspirin to diminish the composed bleeding risk of three different antithrombotic agents. With the same point of view, the POPULAR-AGE (Clopidogrel vs. ticagrelor or prasugrel in patients aged 70 years or older with non-ST-elevation acute coronary syndrome) trial showed that clopidogrel is a favorable alternative for patients above 70 years old as they are considered to be at higher bleeding risk [46].

### 3.3. Prasugrel

As a prodrug like clopidogrel, prasugrel needs metabolic activation and irreversibly inhibits the P2Y_12_ receptor by binding to the same site as ADP. Prasugrel has a faster onset of antiplatelet action compared to clopidogrel due to more efficient metabolic conversion, which requires a single-step hepatic oxidation process to produce an active metabolite with higher in vivo availability [47,48,49], and it has established itself as superior to clopidogrel for the prevention of (recurrent) ischemic events in ACS patients [50,51,52]. As expected, the incidence of major bleeding (including life-threatening and fatal bleeding) was significantly higher in patients treated with prasugrel compared to clopidogrel. Furthermore, a subgroup analysis of TRITON-TIMI 38 showed net harm in prasugrel-treated patients with a prior stroke or transient ischemic attack (TIA), who developed more intracranial bleeding than clopidogrel-treated patients. Therefore, prasugrel therapy contains a caution against its use in patients with a history of TIA or stroke and provisos concerning its use in older and lighter patients [53].

Prasugrel (60 mg loading dose and 10 mg maintenance dose) administration is currently limited to ACS patients whose coronary anatomy has been established, except for STEMI patients undergoing primary PCI. The TRILOGY-ACS (Targeted Platelet Inhibition to Clarify the Optimal Strategy to Medically Manage Acute Coronary Syndrome) [50] and the ACCOAST (Pretreatment with Prasugrel in Non-ST-Segment Elevation Acute Coronary Syndromes) [54] trials analyzed the significance of prasugrel in NSTEMI patients managed with medical therapy alone and as pretreatment in P2Y_12_-naive patients, respectively. In contrast to the TRITON-TIMI 38, the findings of TRILOGY-ACS showed no significant reduction in the composite endpoint of CV death, MI, or stroke at 2.5 years by prasugrel compared with clopidogrel. Pretreatment with prasugrel showed no improvement in the primary endpoint, a composite of CV death, MI, stroke, urgent revascularization, or the need for GPI bailout treatment at seven days, compared with standard administration of prasugrel directly after PCI [50]. Additionally, prasugrel pretreatment was associated with a threefold risk of developing major bleeding and a sixfold risk of developing life-threatening bleeding [54]. Therefore, the results of the ACCOAST trial do not support the administration of prasugrel before angiography in NSTEMI patients.

### 3.4. Ticagrelor

Ticagrelor, unlike clopidogrel and prasugrel, is the first reversibly binding P2Y_12_ receptor inhibitor that does not particularly antagonize ADP binding but impedes concentration-related platelet inhibition by blocking ADP-induced signal transduction [31]. Compared to clopidogrel, ticagrelor has a faster onset of action as it is not a prodrug, does not require metabolic activation, and has a faster offset due to its reversible effects [55]. The phase III PLATO (Ticagrelor vs. Clopidogrel in Patients with Acute Coronary Syndromes) trial showed that ticagrelor reduced the composite endpoint of CV death, MI, and stroke by 16% at one year compared with clopidogrel (HR 0.84, 95% CI 0.77–0.92; *p* < 0.001) in ACS patients, which was primarily driven by a reduction in CV death and MI, at the expense of an increased risk of TIMI-defined non-coronary artery bypass grafting (CABG)-related major bleeding and fatal intracranial bleeding (2.8% vs. 2.2%, *p* = 0.03) [56]. Ticagrelor also proved to be superior to clopidogrel therapy among patients with CV risk factors such as chronic kidney disease [57], diabetes mellitus [58], and ischemic stroke [59].

The 2017 European Society of Cardiology (ESC) guideline on DAPT in CAD gives a class of IB recommendation (indicating that the treatment is recommended) for the administration of either ticagrelor or prasugrel over clopidogrel in ACS patients who undergo PCI and in those who are not at high risk of bleeding [4]. However, limited data compare the efficacy and safety of ticagrelor and prasugrel compared to each other in ACS patients.

The PRAGUE-18 (Prasugrel Versus Ticagrelor in Patients With Acute Myocardial Infarction Treated With Primary Percutaneous Coronary Intervention) [60] was the first head-to-head clinical trial to randomize 1230 STEMI patients to receive either prasugrel or ticagrelor prior to primary PCI. The primary composite outcome was death, re-infarction, urgent TVR, stroke, severe bleeding requiring transfusion, or prolonged hospitalization at seven days. The 30-day follow-up found no significant difference between prasugrel and ticagrelor concerning efficacy or bleeding (odds ratio (OR) 0.98, 95% confidence interval (CI) 0.55–1.73; *p* = 0.939). The 1-year follow-up also showed no significant difference between the two groups (hazard ratio (HR) 1.167, 95% CI 0.742–18.35; *p* = 0.503) [61]. This trial was halted prematurely due to lack of utility and had several limitations, mainly that it was underpowered to show superiority of one agent over another.

A few comparative pharmacodynamic studies have shown little difference between prasugrel and ticagrelor in terms of timing and level of platelet inhibition [62,63,64]. The ISAR-REACT-5 (Intracoronary Stenting and Antithrombotic Regimen: Rapid Early Action for Coronary Treatment 5) [65] study randomized 4018 ACS (STEMI and NSTEMI) patients for whom invasive evaluation was planned and who received either ticagrelor or prasugrel. The primary composite endpoint was death, MI, or stroke, and the principal secondary safety endpoint was bleeding at 1-year follow-up. Based on the results of previous studies regarding the safety and use of prasugrel, patients with a history of stroke, TIA, or intracranial bleeding were excluded. Patients older than 75 years or who weighed less than 60 kg received a daily maintenance dose of 5 mg of prasugrel instead of 10 mg. At the 1-year follow-up, prasugrel therapy was associated with a significant reduction in the primary outcome compared to ticagrelor (6.9% vs. 9.3%; HR 1.36; 95% CI, 1.09 to 1.70; *p* = 0.006), and without a significant difference in bleeding. However, this trial was limited by its open-label design with a modest sample size, and in the intention-to-treat analysis, 32.5% of patients in the ticagrelor group and 30.4% of patients in the prasugrel group were not treated with the assigned drug. Given the inconsistency of these results and the smaller sample size compared with previous trials, a definitive statement of the superiority of prasugrel compared with ticagrelor cannot be made at this time. However, the 2020 ESC-ACS guidelines give a class IIa recommendation (level of evidence B) for using prasugrel over ticagrelor for NSTEMI patients who undergo PCI [17].

## 4. Alternative Approaches to the Administration of Oral P2Y_12_ Receptor Inhibitors

The delayed onset and attenuated antiplatelet effects of oral P2Y_12_ receptor inhibitors, which have been consistently observed in STEMI patients, are mainly attributed to impaired drug absorption resulting in decreased drug availability. Therefore, strategies were developed to improve the bioavailability of orally administered agents and to achieve a faster and more immediate antiplatelet effect, such as crushing or chewing the tablets or increasing the dosing.

The MOJITO (Ticagrelor crushed tablets administration in STEMI patients) trial randomized STEMI patients undergoing primary PCI to receive crushed or integral tablets of ticagrelor. It found an earlier platelet inhibition in favor of crushed ticagrelor (median platelet reactivity units (PRU) was 168 at 1 h after administration of loading dose for crushed ticagrelor vs. 252 for integral tablets, *p* = 0.006) [66]. The strategy of chewed ticagrelor may also have the potential to achieve a faster and stronger platelet inhibition compared to integral or crushed tablets of ticagrelor. This approach was probably suggested due to more enzymatic metabolic degradation of the chewed tablet in the mouth due to the prolonged contact of the drug with the saliva and probably enhanced oral transmucosal absorption of the drug [67,68].

Similar findings were observed in the COMPARE-CRUSH trial using crushed prasugrel in STEMI patients [69]. The administration of crushed prasugrel showed a faster drug absorption according to pharmacokinetic assessments. It reduced the time to effective platelet inhibition as early as 30 min after administration of the loading dose, which persisted up to 4 h compared with the standard integral tablet. However, these findings did not significantly translate into better coronary perfusion pre-PCI.

The improved platelet inhibition with crushed or chewed P2Y_12_ receptor inhibitors was confirmed in a recent meta-analysis [70]. Thus, chewed or crushed P2Y_12_ inhibitors may reduce the time to adequate platelet inhibition but cannot fully bridge the gap.

Another approach was to implement higher loading dosing regimens of ticagrelor and prasugrel to achieve a faster and adequate platelet inhibition. Higher loading doses of ticagrelor (270 mg or 360 mg) compared to the standard loading dose (180 mg) failed to accelerate or enhance platelet inhibition [71]. The levels of high on-treatment platelet reactivity (HPR), which is associated with ischemic events, were still present in the first 2 h after the ticagrelor loading dose, and no significant differences in PRU levels were found across the groups with different loading doses at all time points, except at 1 h with a 270 mg loading dose (*p* = 0.017). In contrast, a higher loading dose of prasugrel (100 mg) resulted in more consistent platelet inhibition compared to the standard loading dose (60 mg) [72]; however, the platelet inhibition was mildly enhanced and was not able to fully bridge the initial gap.

Another alternative approach to achieve optimal platelet inhibition by oral P2Y_12_ receptor inhibitors is to avoid concomitant administration of opioids, as their use is associated with delayed intestinal drug absorption [73]. Although the class of recommendation for administration of opioids has been reduced in the European and American guidelines (class I to IIa recommendation, level of evidence C) on the management of STEMI patients with severe pain [16], opioids are still widely used. Furthermore, gastrointestinal side effects such as nausea and vomiting (throwing up the loading dose of P2Y_12_ receptor inhibitor) are often observed in STEMI patients after receiving opioids [74]. This may lead to reduced bioavailability and a re-loading dose of the P2Y_12_ receptor inhibitors. The ON-TIME 3 (Opioids aNd crushed Ticagrelor In Myocardial infarction Evaluation) trial [75] showed that IV paracetamol, an alternative analgesic, also provided effective pain relief and meanwhile led to significantly higher ticagrelor plasma levels compared to IV fentanyl. Even though the PRU was not significantly lower in STEMI patients treated with IV paracetamol, this study emphasizes the adverse effects of opioids on the bioavailability of oral P2Y_12_ receptor inhibitors.

## 5. Intravenous Agents

### 5.1. Cangrelor

An intravenous P2Y_12_ receptor inhibitor may overcome the inherent limitations of orally administered drugs, such as total dependence on oral administration, which restricts their use in hemodynamically unstable patients who cannot take oral medication, delayed onset of action, and variability in pharmacodynamic responses [73,74].

Cangrelor is the first and only IV P2Y_12_ receptor inhibitor that binds directly to the ADP receptor and induces reversible inhibition like ticagrelor. The drug has been shown to provide rapid and sustained platelet inhibition of more than 90% within 2 min when administered as a bolus followed by infusion. It acts directly on the P2Y_12_ receptor and does not require metabolic conversion in the liver. These characteristics are ideal in STEMI patients and patients with a high thrombotic burden to achieve faster platelet inhibition. Cangrelor is also known for its rapid offset with a short half-life of 3–5 min after discontinuation of the treatment and restores the hemostasis within 30–60 min [23]. Furthermore, no dose adjustment is required in patients with kidney or liver impairment. A condition is that the administration of the loading doses of prasugrel and clopidogrel should be delayed until the end of the cangrelor infusion, as the latter blocks the binding of active metabolites of prasugrel and clopidogrel to the P2Y_12_ receptors.

The role of cangrelor as an adjunctive antiplatelet therapy has been studied in three large-scale trials in the setting of ACS, referred to as the CHAMPION (Cangrelor vs. Standard Therapy to Achieve Optimal Management of Platelet Inhibition) trials, which showed a potential beneficial role of cangrelor in ACS patients undergoing primary PCI [76,77,78]. A pooled analysis of all three CHAMPION trials showed a reduction in peri-procedural ischemic events counterbalanced by a slight increase in minor bleeding [24]. Due to its proven efficacy in preventing ST, the current ESC guideline states that cangrelor can be considered in P2Y_12_-naive patients undergoing primary PCI (class IIb of recommendation, level of evidence A) [71]. A nationwide registry executed in the Netherlands confirmed the safety and feasibility of cangrelor in a broad range of patients, from P2Y_12_-naive patients with stable CAD to STEMI and OHCA patients with a high thrombotic risk who underwent primary PCI [79]. However, the benefit of adding cangrelor to the newer oral P2Y_12_ receptor inhibitors remains unclear, as no phase III trials compared cangrelor to these agents.

### 5.2. Glycoprotein IIb/IIIa Inhibitors

Over the past two decades, GPIs have had varying roles as antiplatelet agents in a wide range of patients undergoing PCI. They prevent platelet aggregation by targeting the final common pathway by competing with fibrinogen and von Willebrand factor (vWF) to bind to activated GPI receptors on the surface of the platelets [80]. The GPIs are administered as a bolus, followed by continuous infusion to achieve an 80–90% platelet inhibition within 10 min. Due to their fast-acting and potent antiplatelet properties, the early administration of GPIs in STEMI patients has improved clinical and angiographic outcomes. However, in most of these studies, patients were pretreated with ticlopidine or clopidogrel and not with the newer P2Y_12_ receptor inhibitors [80,81,82]. Among the three GPI agents (abciximab, eptifibatide, and tirofiban), tirofiban has the most consistent data for prehospital use and is the only one still used. Similar to abciximab and eptifibatide, initial evidence demonstrated the association between earlier use of tirofiban in STEMI patients and improved ST-segment resolution and target vessel perfusion prior to PCI [83]. Most evidence supporting the additional benefit of prehospital administration of tirofiban in STEMI patients is obtained from the ON-TIME (Ongoing Tirofiban In Myocardial Infarction Evaluation) 2 trial [9]. This double-blind, randomized multicenter study demonstrated a significant resolution of ST-segment deviation both pre-and post-PCI among STEMI patients treated with a high dose of prehospital tirofiban on top of standard care compared to patients with standard care alone. Prehospital administration of tirofiban was also associated with significantly reduced rates of death, recurrent MI, urgent TVR, or blinded bailout use of tirofiban at 30 days. Exploratory analyses also demonstrated improvement in initial thrombus burden and initial patency of the IRA in the tirofiban arm [84]. Furthermore, an additional analysis combined pooled data from the open-label phase and the blinded phase of the trial and showed a significant reduction in MACE at 1 year among patients treated with tirofiban compared with placebo.

Despite these findings, the use of GPIs is limited owing to high rates of bleeding complications and the development of alternative strategies with improved safety profiles. The current ESC guideline recommends GPI use only in bailout procedures if there is a no-reflow phenomenon or a thrombotic complication (class IIa recommendation, level of evidence C) [16]

## 6. Subcutaneous Agents

### 6.1. Zalunfiban

So far, GPIs require IV administration of a bolus dose followed by a continuous infusion regulated by an infusion pump, which may further challenge its use in the prehospital setting. Zalunfiban (formerly known as RUC-4) belongs to the GPI family but differs from current GPIs by its subcutaneous route of administration and its capacity to lock the GPI receptors on the surface of the platelets in its inactive conformation [85]. It is a fast-acting (5–15 min) small-molecule inhibitor of the GPI receptor, specifically developed to facilitate prehospital treatment of STEMI at the earliest time to preserve cardiac muscle and minimize the chance of early death. Zalunfiban provides a high grade of platelet aggregation inhibition (≥80%) shortly after administration. Mainly for STEMI patients, early treatment with zalunfiban could be beneficial by improving the initial patency of the IRA and myocardial reperfusion. Thus, improving coronary artery and myocardial microvascular blood flow may decrease infarct size and reduce complications of STEMI.

A Phase I trial evaluated the safety of two different doses (0.05 and 0.075 mg/kg) of zalunfiban in a total of 48 subjects (14 healthy volunteers and 28 patients with stable CAD) [86]. Patients in both groups achieved more than 80% platelet inhibition within 15 min of administration, resolved within 2 h. Concerning the safety profile of zalunfiban, most adverse events were mild, and none led to study drug discontinuation.

A Phase IIa open-label study assessed the pharmacodynamic and pharmacokinetic properties of a single subcutaneous injection of three different doses (0.075, 0.090, and 0.110 mg/kg) of zalunfiban in 27 STEMI patients with planned primary PCI [87]. The primary endpoint of high-grade (≥77%) platelet inhibition at 15 min was achieved in three of eight (dose 0.075 mg/kg), in seven of eight (dose 0.090 mg/kg), and in seven of eight (dose 0.110 mg/kg) patients. Concerning safety, injection site reactions or bruising was observed in a total of one (4%) and eleven (41%) patients; mild access-site hematomas occurred in six (22%); and severe access-site hematomas occurred in two patients (7%). No thrombocytopenia was observed within 72 h post-administration.

The ongoing phase III prospective, double-blind, randomized, placebo-controlled, international multicenter study (ClinicalTrials.gov Identifier: NCT04825743) assesses the efficacy and the safety of a single subcutaneous injection of zalunfiban in STEMI patients in the prehospital setting [29].

The simple administration of zalunfiban, possibly by an autoinjector, makes it a promising approach for managing STEMI at first medical contact, a major unmet need in prehospital care that could potentially improve patient outcomes.

### 6.2. Selatogrel

Selatogrel (formerly ACT-246475) is a reversible and highly selective non-thienopyridine antagonist of the P2Y_12_ receptor, also designed for subcutaneous administration [88].

The early antiplatelet response and safety of subcutaneous selatogrel have been evaluated in two phase II trials among patients with chronic coronary syndrome (CCS) and ACS with planned PCI. The initial study evaluated the pharmacodynamics and pharmacokinetics of selatogrel in 345 CCS patients, who were randomized to receive two different doses of selatogrel (8 mg (n = 114) or 16 mg (n = 115)) or placebo (n = 116) on background oral antiplatelet therapy [89]. At 30 min after selatogrel administration, 89% of patients were responders (defined as PRU < 100) to the 8 mg dose, 90% to the 16 mg dose, and 16% to placebo (*p* < 0.0001). Furthermore, additional platelet inhibition was seen in patients in the selatogrel arm who were already treated with oral P2Y_12_ receptor inhibitors. Concerning the safety of selatogrel, it was confirmed that selatogrel is safe and well-tolerated, with transient dyspnea occurring in 7% of patients (95% CI: 4–11%) overall.

The inhibition of platelet aggregation by selatogrel was also evaluated in ACS patients in another phase II study [90]. A total of 47 patients with ACS (29 with STEMI) were randomized to a single subcutaneous dose of selatogrel of 8 or 16 mg along with standard care. At 30 min after selatogrel administration, 91% of patients were responders (defined as PRU < 100) to the 8 mg dose and 96% to the 16 mg dose. Safety was assessed up to 48 h after selatogrel administration, and selatogrel was well-tolerated without major bleeding complications.

A single subcutaneous dose of selatogrel is thus a promising prehospital treatment for ACS patients as it provides rapid inhibition of platelet aggregation. Selatogrel was found safe and well-tolerated by both ACS and CCS patients; adverse events were mild in severity, and bleeding events were mainly minor in severity.

An ongoing phase III study with subcutaneous selatogrel (ClinicalTrials.gov Identifier: NCT04957719) evaluates this agent’s clinical efficacy when self-administered upon the occurrence of symptoms suggestive of ACS in participants at risk of having a recurrent MI [28].

## 7. Anticoagulation in ACS

Current ESC-ACS guidelines strongly recommend an adjuvant parenteral anticoagulant to antiplatelet therapy during the acute phase of STEMI and NSTEMI, especially during PCI [15,16]. This synergetic approach underscores the importance of thrombin-induced platelet activation and targets other pathways in thrombus generation to mitigate adverse ischemic events [91]. However, appropriate dosing of anticoagulant agents is most important to balance the risk of thrombotic complications and the risk of peri-procedural bleeding. Recommended anticoagulant agents for temporary use in contemporary practice include unfractionated heparin (UFH), enoxaparin, fondaparinux, and bivalirudin. UFH is the anticoagulant of choice in STEMI patients and anticoagulant-naive NSTEMI patients during PCI and has a class I recommendation [92]. However, the evidence is limited (level of evidence C) as no placebo-controlled trial has evaluated the efficacy and safety of UFH in ACS patients undergoing primary PCI.

### 7.1. Enoxaparin

Enoxaparin belongs to the family of low-molecular-weight heparin (LMWH) and has the largest body of clinical data in ACS [92]. It targets a more proximal part of the coagulation pathway by inhibiting activated factor Xa (FXa) and, to a lesser extent, thrombin (factor IIa). Therefore, enoxaparin has the potential for more significant thrombin inhibition as FXa catalyzes thrombin formation compared to UFH. More potential advantages of enoxaparin over UFH are the (1) more reliable dose–effect negating the need for monitoring, (2) bioavailability with either intravenous or subcutaneous administration, and (3) less risk of developing heparin-induced thrombocytopenia (HIT) [93].

The potential benefit of enoxaparin was shown in the ATOLL (Acute Myocardial Infarction Treated with Primary Angioplasty and Intravenous Enoxaparin or Unfractionated Heparin to Lower Ischemic and Bleeding Events at Short- and Long-term Follow-up) trial [94], in which an IV bolus of enoxaparin (0.5 mg/kg) was compared to UFH (50–70 IU/kg with GPIs or 70–100 IU/kg without GPIs) in 910 STEMI patients undergoing primary PCI. Enoxaparin significantly reduced the composite secondary endpoint of mortality, recurrent MI or ACS, and urgent revascularization. Importantly, the safety of enoxaparin and UFH was similar without a significant difference in bleeding endpoints. A per-protocol analysis of the ATOLL trial (87% of the study population) showed the superiority of enoxaparin over UFH in reducing death, ischemic endpoints, and major bleeding. Based on these results, current ESC guidelines recommend enoxaparin to be considered in STEMI patients as an alternative for UFH (class IIa of recommendation, level of evidence A). Notably, this study was conducted before the introduction of ticagrelor, and more than 75% of all patients received GPIs concomitantly, so the findings of this study may no longer apply to current practice.

The more recent PENNI PCI (Pharmacodynamic Effects of a 6-Hour Regimen of Enoxaparin in Patients Undergoing Primary Percutaneous Coronary Intervention) trial [95] measured anti-FXa levels at four different time points to evaluate the pharmacodynamic effects of the enoxaparin regimen (bolus followed by infusion) in 20 STEMI patients during primary PCI. Enoxaparin resulted in sustained anti-Xa levels during infusion without bleeding complications. Therefore, enoxaparin may still be an attractive alternative to UFH in primary PCI, particularly in opiate-treated patients with delayed platelet inhibition by oral P2Y_12_ receptor inhibitors.

### 7.2. Fondaparinux

Fondaparinux, a synthetic pentasaccharide, selectively binds to antithrombin, which leads to rapid and irreversible inhibition of factor Xa [96]. Although this parenteral agent exclusively targets factor Xa and none of the other coagulation factors, it still effectively inhibits thrombin generation. Furthermore, there is no need for monitoring due to its predictable dose–effect after subcutaneous injection.

The OASIS-5 (Comparison of Fondaparinux and Enoxaparin in Acute Coronary Syndromes) trial [97] determined whether fondaparinux would preserve the anti-ischemic benefits of enoxaparin in NSTEMI patients. It succeeded in showing fondaparinux to be non-inferior, irrespective of management strategy, to enoxaparin in preventing the occurrence of death, MI, or refractory ischemia at nine days among high-risk patients with unstable angina or NSTEMI. Mainly, fondaparinux significantly reduced bleeding events, associated with substantially lower long-term (3 to 6 months) mortality, MI, and stroke.

The beneficial effect of fondaparinux in STEMI patients was assessed in the OASIS-6 (Organization for the Assessment of Strategies for Ischemic Syndromes) trial [98], whereby patients received either fondaparinux on top of standard therapy or usual care. The study showed the superiority of fondaparinux to standard therapy (i.e., placebo or UFH) in terms of antithrombotic efficacy (death and re-infarction), with no increase in bleeding risk. STEMI patients who did not receive reperfusion therapy had the beneficial effects of fondaparinux, which does not apply to the patients undergoing primary PCI. Based on these latter findings, the current ESC guidelines do not recommend using fondaparinux during primary PCI [16].

Overall, a single dosage of fondaparinux (2.5 mg QD) preserved or even improved the anti-ischemic benefits of traditional anticoagulant agents across the spectrum of ACS patients without increasing bleeding risk.

### 7.3. Bivalirudin

Bivalirudin is a reversible and direct thrombin inhibitor with predictable anticoagulant effects. Evidence of its beneficial effect in STEMI stems primarily from the HORIZONS-AMI (Bivalirudin during Primary PCI in Acute Myocardial Infarction) trial [99], in which STEMI patients undergoing primary PCI were assigned to either bivalirudin or heparin plus a GPI. A significant reduction in death from cardiac and non-cardiac causes and major bleeding at 30 days was observed with bivalirudin compared to heparin plus a GPI, with similar rates of major adverse CV events among the two groups. The mortality advantage of bivalirudin owed to the significantly lower rates of bleeding, specifically iatrogenic hemorrhagic complications during PCI. Moreover, bivalirudin reduced the occurrence of severe thrombocytopenia, which has also been strongly associated with death among patients with STEMI and PCI.

Nonetheless, a meta-analysis of five dedicated randomized controlled trials showed no mortality reduction with bivalirudin compared to UFH in STEMI patients, and most importantly, bivalirudin was associated with an increased risk of acute ST. On the other hand, the findings of reduced bleeding risk by bivalirudin were aligned with the results of the HORIZONS-AMI trial. The more recent MATRIX (Bivalirudin or Unfractionated Heparin in Acute Coronary Syndromes) trial [100] also showed no reduction in mortality and ischemic endpoints by bivalirudin compared to UFH in ACS (56% STEMI) patients undergoing PCI. However, a post hoc analysis of the MATRIX trial implied an association between a prolonged bivalirudin infusion after PCI and the lowest risk of ischemic and bleeding events.

The current practice guidelines recommend bivalirudin as an alternative anticoagulant agent for UFH during primary PCI in patients with heparin-induced thrombocytopenia and at higher risk of bleeding.

### 7.4. Non-Vitamin-K-Antagonist Oral Anticoagulants (NOACs)

Combining DAPT and an oral anticoagulant is associated with a two- to threefold higher risk of bleeding without a significant reduction in ischemic risk in atrial fibrillation patients with ACS undergoing PCI [101,102]. Hence, the introduction of the novel oral anticoagulants (NOACs) inspired curiosity about whether the concomitant use of these agents with DAPT can still add a benefit to mitigate ischemic risk without increasing bleeding risk in ACS patients. Several studies have assessed strategies to answer this question.

The RE-DEEM (Dabigatran vs. placebo in patients with acute coronary syndromes on dual antiplatelet therapy) is a randomized, double-blind, phase II trialthat evaluated the efficacy and safety of various doses of dabigatran on top of DAPT compared to placebo in NSTEMI and STEMI patients [103]. It showed a dose-dependent increase in major and clinically relevant minor bleeding without significant reduction in ischemic endpoints in the dabigatran group.

The phase II APPRAISE (Apixaban, an Oral, Direct, Selective Factor Xa Inhibitor, in Combination With Antiplatelet Therapy After Acute Coronary Syndrome) trial [104] assessed the efficacy and safety of four different doses of apixaban (ranging from 2.5 mg–10 mg BID and 10 mg–20 mg QD) in patients with NSTEMI or recent STEMI on top of aspirin or aspirin plus clopidogrel compared to placebo. The addition of apixaban to antiplatelet therapy showed a dose-related increase in bleeding and a trend toward reducing ischemic events in ACS patients. The phase III APPRAISE-2 (Apixaban with Antiplatelet Therapy after Acute Coronary Syndrome) trial compared apixaban 5 mg BID (or 2.5 mg BID in case of renal impairment) with placebo in patients with recent ACS and at least two additional risk factors for recurrent ischemic events. The trial was terminated prematurely as the concomitant use of apixaban and antiplatelet therapy (aspirin or aspirin plus clopidogrel) significantly increased TIMI major bleeding without significantly reducing ischemic endpoints.

Finally, the dose-finding phase II ATLAS ACS 2-TIMI 46 (Rivaroxaban vs. placebo in patients with acute coronary syndromes) [105] and phase III ATLAS ACS-2-TIMI 51 (Rivaroxaban in Patients with a Recent Acute Coronary Syndrome) [106] trials studied the efficacy and safety of rivaroxaban in ACS setting. The phase III trial included 15,526 patients with recent ACS who were randomized in a 1:1:1 method to receive either rivaroxaban 2.5 mg BID or 5 mg BID or placebo on top of standard antiplatelet therapy (aspirin or aspirin plus P2Y_12_ receptor inhibitor). Rivaroxaban in addition to antiplatelet agents significantly reduced the composite of CV death, MI, or stroke (8.9% vs. 10.7%, *p* = 0.008), but at a cost of higher rates of non-CABG-related major bleeding (2.1% vs. 0.6%, *p* < 0.001) and intracranial bleeding (0.6% vs. 0.2%, *p* = 0.009).

Overall, the addition of an anticoagulant to contemporary ACS treatment with aspirin and clopidogrel is associated with higher bleeding risk. So far, rivaroxaban is the only NOAC that has been shown to reduce mortality and ischemic events, although these benefits are also counterbalanced by higher rates of major bleeding. Current ESC guidelines suggest considering (class IIb recommendation) low-dose rivaroxaban (2.5 mg BID) on top of DAPT (aspirin and clopidogrel) if ischemic risk is expected to be higher than bleeding risk in patients with recent ACS. However, the addition of rivaroxaban on top of ticagrelor or prasugrel has not been studied yet.

### 7.5. Factor X1(a) Inhibitors

A potential new anticoagulant that may prevent ischemic events while maintaining lower bleeding risk is the factor X1(a) inhibitor. Factor X1 (eleven) is a plasma glycoprotein that circulates in the inactive form and becomes activated by factor XIIa to factor X1a. The activated factor XIa in turn acts in the intrinsic pathway and is responsible for the amplification of thrombin generation.

The potential role of factor X1 was described in patients with factor X1 deficiency. These patients with low levels of factor X1 seem to be protected against thrombosis and consequently have a significantly lower risk of ischemic stroke than the general population [107]. These findings generated the hypothesis that inhibiting either factor X1 or the active form factor X1a may be a promising approach to prevent adverse ischemic events in ACS patients, hence not increasing the risk of bleeding.

The recently completed PACIFIC-AMI (Proper Dosing and Safety of the Oral FXIa Inhibitor BAY 2433334 in Patients Following an Acute Heart Attack) was a multicenter, randomized, placebo-controlled, double-blind, parallel-group, dose-finding phase II trial, which studied the efficacy and safety of the oral FXIa inhibitor asundexian (BAY 2433334) on top of oral DAPT in patients following an acute MI. If the trial results show an improved safety profile compared to the conventional anticoagulants, the factor X1(a) inhibitor could be a promising option in ACS patients and expands treatment possibilities to an even wider group of CV patients.

## 8. Triple Antithrombotic Therapy in ACS

Incident atrial fibrillation (AF) occurs in approximately 20% of ACS patients who undergo PCI [108]. These patients are at a higher risk of ischemic stroke (IS) and in-hospital mortality than ACS patients without AF [109,110,111]. Furthermore, AF can be a risk factor for (recurrent) ACS as they share the same cardiometabolic risk profiles. ACS patients with AF have an indication for triple antithrombotic therapy with an oral anticoagulant (vitamin K antagonist (VKA) or NOAC) and a dual therapy at least 1 year post-ACS.

However, triple therapy after ACS is associated with a higher risk of bleeding [112,113,114], much higher in patients treated with more potent P2Y_12_ receptor inhibitors such as prasugrel and ticagrelor [115].

Warfarin is the most and most thoroughly studied VKA in ACS patients. The WOEST (What Is the Optimal Antiplatelet and Anticoagulant Therapy in Patients with Oral Anticoagulation and Coronary Stenting) trial [105] showed a lower bleeding risk in AF patients with recent ACS treated with warfarin and clopidogrel compared to warfarin-based triple therapy. However, this study suggests triple therapy as the effective regimen to balance the ischemic and bleeding risk in such patients. However, this study was not powered to show a difference in the ischemic endpoint of ST.

With the increasing use of NOACs in AF patients in recent years, various studies have compared NOAC-based triple treatment to warfarin-based triple treatment in AF patients with ACS. The PIONEER AF-PCI (Open-Label, Randomized, Controlled, Multicenter Study Exploring Two Treatment Strategies of Rivaroxaban and a Dose-Adjusted Oral Vitamin K Antagonist Treatment Strategy in Subjects with Atrial Fibrillation Who Undergo Percutaneous Coronary Intervention) trial [116] compared three different antithrombotic regimens that included warfarin and the NOAC rivaroxaban: (1) low-dose rivaroxaban (15 mg QD or 10 mg QD if creatinine clearance 30–50 cc/min) + a P2Y_12_ receptor inhibitor; (2) very low-dose rivaroxaban (2.5 mg BID) + DAPT; and (3) warfarin + DAPT. The combination of rivaroxaban and a P2Y_12_ receptor inhibitor was associated with a lower rate of clinically significant bleeding than warfarin and a P2Y_12_ receptor inhibitor. However, TIMI-defined major bleeding and MACE were similar between the three groups, except for all-cause mortality and rehospitalization, which was less frequent in rivaroxaban-based therapy. The RE-DUAL PCI (Randomized Evaluation of Dual Antithrombotic Therapy with Dabigatran vs. Triple Therapy with Warfarin in Patients with Nonvalvular Atrial Fibrillation Undergoing Percutaneous Coronary Intervention) trial [117] showed similar results in bleeding events in patients treated with dabigatran and a P2Y12 receptor inhibitor compared to warfarin-based triple therapy. Both studies have confirmed that NOACs result in less bleeding than warfarin in high-risk patients with AF for whom PCI is required.

The AUGUSTUS (Antithrombotic Therapy after Acute Coronary Syndrome or PCI in Atrial Fibrillation) trial [118] evaluated the independent effects of anticoagulants apixaban and warfarin with different antithrombotic regimens in AF patients after ACS or PCI. The combination of a P2Y_12_ receptor inhibitor and apixaban without aspirin led to fewer bleeding events than apixaban-based triple therapy, warfarin-based triple therapy, and warfarin alone with a P2Y_12_ receptor inhibitor. Aspirin led to higher bleeding and was associated with fewer ischemic events only 30 days after ACS or PCI.

Therefore, for ACS patients with AF and an indication for oral anticoagulation, the new default guideline recommendation is seven days of triple therapy, and after that, dual therapy with clopidogrel and a NOAC for at least 12 months. For those patients at higher ischemic risk, triple antithrombotic therapy can be extended up to 4 weeks.

## 9. Conclusions

Much progress in antithrombotic therapy has been made in recent years and has led to changes in the recommendations of current guidelines for ACS. The totality of the evidence of different antithrombotic regimens acquired from clinical trials is the foundation for these guidelines. However, it is essential to realize that this evidence reflects population-level data. A real-world individual patient cannot always fit into the inclusion criteria of trials, as sociodemographic, clinical, or other relevant characteristics require special consideration.

Newly introduced antithrombotic agents in clinical practice are cangrelor, enoxaparin in STEMI, and rivaroxaban as an adjunctive in ACS. Other potentially interesting drugs are currently being developed, which include several novel potent antiplatelet drugs targeting alternative pathways. Furthermore, the development of an FXI(a) inhibitor seems promising, with the potential of reducing thrombus formation with only minimal effect on bleeding. Hence, there is a glimpse of several promising new antithrombotic drugs on the horizon. Clinical trials must further prove their efficacy and applicability in the ACS setting.

## Figures and Tables

**Figure 1 jcm-11-05605-f001:**
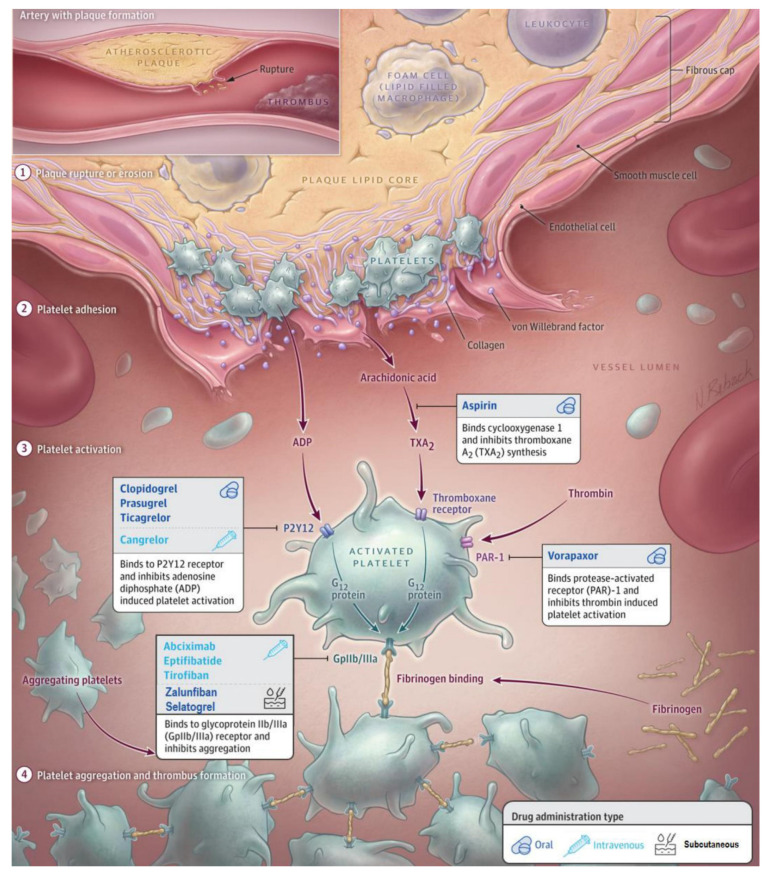
An overview of established and newer antithrombotic agents with different administration types in the context of acute coronary syndrome.

**Table 1 jcm-11-05605-t001:** Major characteristics of traditional and newer antithrombotic agents.

Antithrombotic Agents	Route of Administration	Mechanism of Action	Onset of Action	Peak Effect	Duration of Action	Recommendations Pre- and Post-PCI [16,17]	Trials of Prehospital Administration
ASPIRIN	Oral, IV	Acetylation of platelet cyclooxygenase	30–60 min	1–2 h	10 d	Class I, LOE B	-
** *P2Y12 INHIBITORS* **							
CLOPIDOGREL	Oral	Binding to the P2Y_12_ components of ADP receptor	2 h	6 h	5 d	Class I, LOE A	Zeymer et al. [18]
TICAGRELOR	Oral		30 min	2 h	3 d	Class I, LOE A	Montalescot et al. [19]
PRASUGREL	Oral		<30 min	4 h	5–9 d	Class I, LOE A	Vlachojannis et al. [20]
CANGRELOR	IV		2 min	30 min	1 h	Class IIb, LOE A	-
SELATOGREL	SC		15 min	30–60 min *	4–12 h	-	NCT04957719 ^†^
** *GP IIB/IIIA INHIBITORS* **							
ABCIXIMAB	IV	Binding to GP IIb/IIIa receptors	<10 min	30 min	3–5 h ^a^	Class IIa, LOE C *	Ohlmann et al. [21]
TIROFIBAN	IV		<10 min	30 min	4–8 h	Class IIa, LOE C *	van t Hof et al. [8]
ZALUNFIBAN	SC		<15 min	30–60 min	2–4 h ^b^	-	NCT04825743 ^†^
** *ANTICOAGULANT AGENTS* **							
ENOXAPARIN	IV/SC	ATIII-mediated selective inhibition of FXa	<2 min	3–5 h	12 h	Class IIa, LOE A	Labèque et al. [22]
FONDAPARINUX	IV/SC	Inhibiton of FXa	<2 min	2 h	72 h	Class III, LOE B	-
BIVALIRUDIN	IV	Inhibtion of thrombin	<2 min	<2 min	1 h	Class IIa, LOE A	Jacquemin et al. [23]
NOAC	Oral	Inhibiton of FXa or thrombin	<30 min	2–4 h	24 h	Class IIb, LOE B **	-

Abbreviations: IV, intravenous; SC, subcutaneous; GP, glycoprotein; AT, antithrombin; FXa, factor Xa; h, hour; d, days; LOE, level of evidence. ^a^ Platelet-bound antibody detected in the circulation up to 15 days after administration; ^b^ Dose-dependent; * Bailout use; ** Low-dose rivaroxaban as maintenance therapy; ^†^ Ongoing.

## Data Availability

Not applicable.

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
