# Peer review of "Current and Future Insights for Optimizing Antithrombotic Therapy to Reduce the Burden of Cardiovascular Ischemic Events in Patients with Acute Coronary Syndrome"

_jcm, 2022, doi:10.3390/jcm11195605_

Round 1

Reviewer 1 Report

The pharmacological treatment strategies of acute coronary syndrome (ACS) in recent years are constantly evolving to develop more potent antithrombotic agents, as reflected by the introduction of more novel P2Y12 receptor inhibitors and anticoagulants to reduce the ischemic risk among ACS patients. Despite the substantial improvements in the current antithrombotic regimen, a noticeable number of ACS patients continue to experience ischemic events. Effective ischemic risk reduction while balancing bleeding risk remains a clinical challenge. This updated review discusses the currently approved and widely used antithrombotic agents and explores newer antithrombotic treatment strategies under development for the initial phase of an ACS.

The article is well written and brings new knowledge to the topic of ACS. Nevertheless, there is no reference to extremely important publications highlighting the importance of supplemental statins in relation to DOAC.

For example, the studies by Wańkowicz et al and Yang et al noted:

both non-modifiable and modifiable AF risk factors are also recognized risk factors for IS; and AF patients treated with VKAs and who had therapeutic INR levels, as well as AF patients treated with NOACs, have a higher prevalence of thrombotic, proatherogenic, and pro-inflammatory risk factors. It was observed that the presence of cardioembolic risk factors was independently associated with left atrial volume index, persistent atrial fibrillation, heart failure, and body mass index. In contrast, the presence of non-cardioembolic risk factors was independently associated with coronary artery calcium score, hypertension, diabetes, and age. This indicates that statins may be the ideal candidates to complement the anticoagulation effects of VKAs and NOACs, with their broad-spectrum positive clinical effect on thrombotic, proatherogenic, and proinflammatory risk factors. However, cardiologists do not recommend the routine use of statins in all patients with atrial fibrillation as a primary prophylaxis for ischemic stroke, rather relying solely on the prescription of older or new-generation oral anticoagulants. Neurologists recommend statins only for the prevention of atherosclerotic strokes.

In my opinion, this could add value to this review.

Author Response

Dear reviewer,

We would like to thank you for your review and your worthy suggestions. 

Please kindly see the attached with our response. 

Yours sincerely,

Abi Selvarajah, on behalf of dr. R.S. Hermanides 

Reviewer 2 Report

In the proposed review the authors thoroughly discuss not just widely accepted but also novel treatment options for patients with ACS. Even though they represent updated scientific data in readable and easy to understand manner, I strongly suggest the implementation of at least one table with the most important facts in the manuscript (e.g. current treatment options: representatives, mechanism of actions, ways of application, benefits, risk, recommendations...). Additionally, as optimization of the treatment has been underlined in the title/manuscript, please discuss the importance of personalizing treatment decisions. Namely, randomized clinical trials cited in the manuscript typically enroll highly selected patient populations, and the most vulnerable groups continue to be underrepresented in contemporary ACS clinical trials. Maybe some non-randomized observational evidence can also be useful and could find its place in the presented work. Moreover, triple therapy has not been mentioned in the presented work, even though it is of great importance for the patients  with AF and recent ACS/PCI. it is highly advisable for the authors to mention this in the proposed work.     

Author Response

(The authors gave the same response as above.)

Round 2

Reviewer 1 Report

The article is well written and brings new knowledge to the topic of ACS. Nevertheless, there is no reference to extremely important publications highlighting the importance of supplemental statins in relation to DOAC.

For example, the studies by Wańkowicz et al and Yang et al noted:

both non-modifiable and modifiable AF risk factors are also recognized risk factors for IS; and AF patients treated with VKAs and who had therapeutic INR levels, as well as AF patients treated with NOACs, have a higher prevalence of thrombotic, proatherogenic, and pro-inflammatory risk factors. It was observed that the presence of cardioembolic risk factors was independently associated with left atrial volume index, persistent atrial fibrillation, heart failure, and body mass index. In contrast, the presence of non-cardioembolic risk factors was independently associated with coronary artery calcium score, hypertension, diabetes, and age. This indicates that statins may be the ideal candidates to complement the anticoagulation effects of VKAs and NOACs, with their broad-spectrum positive clinical effect on thrombotic, proatherogenic, and proinflammatory risk factors. However, cardiologists do not recommend the routine use of statins in all patients with atrial fibrillation as a primary prophylaxis for ischemic stroke, rather relying solely on the prescription of older or new-generation oral anticoagulants. Neurologists recommend statins only for the prevention of atherosclerotic strokes.

In my opinion, this could add value to this review.

Author Response

Response to reviewer 1: Thank you very much for your positive comments and your suggestions. As you mentioned, concomitant statin therapy is an important treatment strategy in ACS patients. Though, this review aims to provide the most up-to-date information on the current and newer antithrombotic regimen and to emphasize the importance and significance of an evolving pharmacological field about the ACS. As the main focus of this review and this special issue of JCM was antithrombotic therapy in ACS patients, we solely focused on the antithrombotic regimen in ACS. The topic of the importance of other concomitant drugs, including statins, was excluded to keep the focus and to prevent a very long review.

We also discussed the addition of statin therapy in this review with the guest editor, but he also felt that this topic did not fully fit into the theme of the review and the special issue. For this reason, our humble request is to accept our opinion and view. Hopefully, we will receive a positive response.  
